# Assessing Quality of Life with Community Dwelling Elderly Adults: A Mass Survey in Taiwan

**DOI:** 10.3390/ijerph192214621

**Published:** 2022-11-08

**Authors:** Ying-Chen Chi, Chen-Long Wu, Hsiang-Te Liu

**Affiliations:** 1Department of Healthcare Information and Management, Ming Chuan University, 5 De Ming Rd., Gui Shan District, Taoyuan City 333, Taiwan; 2Department of Pediatrics, Taipei City Hospital, No.145, Zhengzhou Rd., Datong Dist., Taipei City 103, Taiwan; 3Department of Public Affairs and Administration, Ming Chuan University, 5 De Ming Rd., Gui Shan District, Taoyuan City 333, Taiwan

**Keywords:** quality of life, WHOQOL-BREF questionnaire Taiwan version, community dwelling elderly adults, physical health, psychological health, social relationships, environmental factors

## Abstract

Background: With the growing life expectancy for older adults, this study aims to examine the correlation among sociodemographic characteristics and the combined effect of QoL-related domains including physical health, psychological health, social relationships, and environmental factors with the overall QoL level of older adults in Taiwan. Methods: The WHOQOL-BREF Taiwanese Version questionnaire was adopted and conducted using a randomized telephone interview system from community household elders. In total, 1078 participants aged 65 years and older were recruited. A multiple regression model was used to examine the statistical significance between the overall QoL score as the dependent variable and the sociodemographic characteristics, and 26 items of QoL-related questionnaires as the independent variables. Results: Categories including female, aged 85 years and above, higher education level, and better financial situation had significantly higher overall QoL level. Except the physical health domain and six items, the correlations among all other domains and their including items of questionnaires with overall QoL level were significant. Conclusion: The Taiwanese WHOQOL-BREF questionnaire can be used to examine the overall QoL level of elders in Taiwan. Nevertheless, the robust systems of universal health care and long-term care in Taiwan may have led to the no significance of the six items.

## 1. Introduction

The World Health Organization (WHO) predicted that between 2015 and 2050, the proportion of people aged 60 years and older will approximately double from 12% to 22% [1,2,3]. Consequently, all countries face important challenges in ensuring that their health and social systems are prepared for such a demographic change [1,2,3]. Taiwan is no exception: the proportion of aged 65 years and older population is expected to increase from 15.3% in 2019 to 36.6% in 2050 [4].

Aging is a continuous process of change in mental and physical health that generally leads to the loss of functional capabilities and ability to perform daily activities [5]. Thus, researchers have increasingly investigated to the measurement of QoL in older adults [1,5,6].

The WHO defined QoL as a person’s perception of their capacity in life and the value that they bring to society [2,6], along with their goals, anticipation, and attention regarding the future [7]. QoL is related to an individual’s present state of and ability to maintain their physical and mental health. Subsequently, QoL also encompasses a person’s level of autonomy, social relationships, and environmental support [8,9].

With regard to saving on long-term care expenditures, it is essential to support the long-term QoL of independent older adults and older adults living in care communities [10]. Previously, several instruments for investigating QoL have been useful for assessing older adults [11]. To evaluate QoL-improving activities for older adults worldwide, the WHO Quality of Life (WHOQOL) group developed a brief QoL assessment, “the World Health Organization Quality of Life, brief version (WHOQOL-BREF),” which comprises 24 items falling under the four domains of physical, psychological, social relationship, and environmental factors [3,6].

Over the past two decades, individual QoL has become more valuable and appreciated in healthcare systems, particularly emphasizing physical health [12,13,14], psychological health [6,15], social relationships, and environmental support [16,17,18,19]. Furthermore, QoL among elderly population has become an inevitable issue and must be addressed under the ageing society worldwide. Nevertheless, only limited studies have demonstrated the effect of the above multiple domains on QoL, especially in elderly populations. In particular, Lin’s study has pointed out that the Taiwanese cultural perception has a societal root in Confucianism (respect for the aged and wise), which has highlighted the cultural value of filial obligation and respect to elders and thus led the long-term care policy making based under the concepts of active aging and aging in place to meet the needs of the aging population in Taiwan. Meanwhile, the well-established medical care system and economic safety condition in Taiwanese society has been revealed in the past two decades [20]. Therefore, the present study aims to examine the impact of physical health, psychological health, social relationships, and environmental factors on QoL of older adults in Taiwan. Sociodemographic characteristics were also accounted for, and data were gathered through a telephone questionnaire survey. We believe that more attention should be paid to the aforementioned factors and their effect on the daily lives of older adults.

The impact effects of the factors reflected in the assessment scale items differ between geographical regions. This study adopted the Taiwanese version of WHOQOL-BREF, which comprises 26 items affecting QoL [21], and explored the correlation between results from the four domains and the overall QoL among older adults in Taiwan. Past studies on QoL have extensively explored differences in quality of life between different age groups and patients. However, there are not many studies on the validity of the four domains of the WHOQOL-BREF questionnaire by means of confirmatory factor analysis. There are still not many studies using the four domains and all facets to predict overall QoL perception in the past. This study can compare the impact of different domains and facets on overall QoL perception of elderly people through standardized coefficients and significance levels. Considering its own specificity of cultural, social, environmental, and medical care system in Taiwan, it is worth exploring whether the scale of WHOQOL-BREF questionnaires of the older adults in Taiwan can predict their overall QoL outcome in this era of a rapidly aging society.

Past studies have found that cultural variables are the most important explanatory factors for QoL in older adults. They found significant differences in QoL between older adults in developed and developing countries [22]. A QoL study of older adults found that individualistic (values referred to as power, achievement, hedonism, stimulation, and self-direction), collectivistic (benevolence, tradition, and conformity) or mixed (security and universalism) cultural differences affect evaluation of QoL [23]. Therefore, this study suggests that there will be cultural differences in QoL application among the elderly in Taiwan.

Our hypotheses are as follows:

**H1.** 
*Sociodemographic characteristics, including gender, age, education level, living arrangement, and financial situation, will predict overall QoL in older adults.*


**H2.** 
*Physical health, psychological health, social relationships, and environmental factors will predict overall QoL.*


**H3.** 
*There will be some cultural differences in the application of the WHOQOL-BREF questionnaire to the elderly in Taiwan.*


## 2. Materials and Methods

### 2.1. Sampling

For this study, a randomized cross-sectional survey was conducted. Data were obtained through a computer-assisted telephone interview system [24] over 2 months (October–December 2020). This study used standardized interviewing to collect raw data. The advantages of standardized interviewing are that it can reduce bias; increase credibility, reliability, and validity; and that it is simple, cost-effective, and efficient. However, standardized interviewing also has some disadvantages: it is difficult to establish a rapport between the interviewer and the participant; the interview is relatively inflexible; and the scope of the interview is very limited. The commissioned survey center has some interview methods to reduce the shortcomings of standardized interviewing. They train interviewers to interview in a friendly way, reducing the interviewee’s sense of strangeness and stress. They train interviewers to avoid bluntly speaking out the questionnaire and allow respondents some flexibility in answering.

Telephone interviews were conducted by trained hirelings of a great Taiwanese opinion polling company, all of whom were well-versed in the local language (i.e., Mandarin or Taiwanese). The telephone survey used a multistage stratified random sampling technique to randomly dial telephone numbers stratified by local governments in Taiwan. Interviewers screen older adults over the age of 65 and those without cognitive impairment for telephone interviews. Of the 1339 elderly people who received a call from the survey center, 261 declined to be interviewed. Finally, a valid sample of 1078 participants was obtained, and the percentage of successful telephone interviews was 80.5%. The estimated error was approximately ±2.99% with a 95% confidence level. This study used G*Power software (latest ver. 3.1.9.7; Heinrich-Heine-Universität Düsseldorf, Düsseldorf, Germany) to calculate sample size and power analysis. The valid sample size exceeded the requisite minimum sample size of 179, which was determined using G*Power software at α error probability = 0.05 and power (1 − β err prob) = 0.90 [25] in this study. In this study, the first 25% of the samples received and the last 25% received were divided into two groups. Independent sample t-test was performed for physical health, psychological health, social relationships, and environmental factors. The t values of physical health, psychological health, social relationships, and environmental factors were: 0.17, 0.76, 0.66, 0.71. The significant levels of the four domains were 0.862, 0.450, 0.512, and 0.478, which were higher than 0.05. It is confirmed that there is no problem of nonresponse bias in the sample collection of this study. There was no significant difference between those who participated and those who did not.

This study was approved by the Research Ethics Committee of the National Taiwan University (IRB No. 201912ES054). This study followed relevant regulations on human research to ensure that this article complies with the principles of medical ethics.

### 2.2. Measures

The questionnaire inquired into overall QoL (the dependent variable), sociodemographic characteristics, and items under the QoL domains (physical health, psychological health, social relationships, and environmental factors, which were the independent variables). All responses were given on a 5-point Likert scale (1 = dissatisfaction, no need, or not enough, 5 = satisfaction, need, and enough). An instructor reviewed the questionnaire for validity and reliability, and the questionnaire was tested and revised before it was given to the participants. The independent variables for evaluating the outcomes were as follows:(1)Sociodemographic: gender, age, education level, living arrangement, and financial situation.

The gender variable distinguishes between men and women. The age variable is divided into three categories: 65 to 74 years old, 75 to 84 years old, and 85 years old and above. Education level variables are divided into illiterate, elementary school, secondary school, senior high school, university, and master’s degree or above. Living arrangement variables are divided into living alone, living with spouse, and living with regular family members.

The options of the financial situation in this study were modified from the classi-fication of the Survey on the Status of the Elderly in Taiwan: well off, roughly enough, slightly difficult, and quite difficult. This study combined “slightly difficult” and “quite difficult” into the “financially struggling” options [26].(2)The Taiwanese version of the WHOQOL-BREF divided into four domains: physical health (7 items), psychological health (6 items), social relationships (4 items), and environmental factors (9 items).

### 2.3. Statistical Analysis

#### 2.3.1. Reliability and Validity

In this study, a confirmatory factor analysis was used for evaluating all items and for confirming the reliability and validity of the four domains. The model fit index was used to confirm the model fit and overall construct validity. Confirmatory factor analysis provides many reliability and validity indicators and is widely used in testing measurement scales.

With regard to absolute model fit, the chi-squared value of the conceptual model was 396, and the degrees of freedom was 293. The model GFI (goodness-of-fit index) was 0.99, which was greater than the requisite minimum of 0.90. The SRMR (standardized root-mean-square residual) was 0.04, which was lower than the requisite maximum of 0.05. About comparative model fit, NNFI (non-normed fit index) = 0.99, NFI (normed fit index) = 0.98, CFI (comparative fit index) = 0.99, and IFI (incremental fit index) = 0.99, all of which were greater than the requisite minimum of 0.90. Regarding parsimonious fit, PNFI = 0.88 and PGFI (parsimony goodness-of-fit index) = 0.77, both of which were greater than the requisite minimum of 0.50. Therefore, the conceptual model had good fit and the study had good construct validity [27].

The factor loading λ values of “dependence on medicinal substances and medical aids”, “negative feelings”, “sexual activity”, “physical environment”, “health and social care: accessibility and quality”, and “food availability” were 0.48, 0.46, 0.48, 0.47, 0.48, and 0.43, respectively, which were close to the requisite minimum of 0.5-the threshold recommended by Hair et al. [28,29]. The “food availability” item had the lowest factor loading and was a new item for the Taiwanese version of the questionnaire. The loadings of items from all other domains ranged from 0.53 to 0.82, which were both higher than the requisite minimum of 0.5. Therefore, the items of each domain were reliable. The loading Z values of all items were significant, which indicated construct and convergence validity.

Furthermore, the latent domain of this study had good internal consistency: the CR values of this study ranged from 0.70 to 0.85, all of which were greater than the requisite minimum of 0.7 recommended by Hair et al. [28,30] (Table 1).

#### 2.3.2. Analytical Strategies

We first presented the sociodemographic characteristics of the study participants. After analysis of reliability and validity, all items were consolidated for path analysis. Since all factor loading values of the 4 domains are higher than 0.4, this study summed the facets scores of all domains separately for path analysis. This article selected item 1 of the WHOQOL-BREF, “Overall, how do you evaluate your QoL (overall QoL),” as a dependent variable. In this study, the tolerance value was between 0.54 and 0.98 (greater than 0.1); the variance inflation factor was between 1.02 and 1.83 (less than 10), which indicated no problems with collinearity [31]. This paper used SPSS 26.0 software to perform multiple regression analysis between independent variables and dependent variable. The level of significance was set at 0.05.

## 3. Results

### 3.1. Sample Description

With regard to the sociodemographic of the older adult study sample, men accounted for 39.0% of participants. The percentages of individuals with illiteracy and with elementary school, secondary school, senior high school, university, and graduate qualifications were 8.6%, 30.7%, 10.5%, 20.2%, 26.5%, and 3.4%, respectively. Regarding living arrangements, older adults living alone, living with spouse, and living with other family members accounted for 11.6%, 26.1%, and 61.8% of the sample, respectively. Regarding financial situation, the older adults who were well-off, roughly enough, and financially struggling constituted 7.1%, 79.5%, and 13.4% of the sample, respectively (Table 2).

For the population variables, this study transformed the nominal scale into dummy variables and added them to the regression equation. With men as the reference group, the QoL of women was relatively high. The standardized coefficient of women was 0.12, which was statistically significant (Table 3).

Taking 65 to 74 years as the reference group, the QoL of older adults who were 85 years and older was relatively high. The standardized coefficient for older adults 85 years and older was 0.18, which was statistically significant (Table 3).

In this paper, the older adults who were not literate were used as the reference group. The QoL of older adults with an education level above elementary school was higher than illiterate group. The standardized coefficients of older adults above elementary school were 0.30, 0.27, 0.39, 0.30, and 0.86, all of which were statistically significant. The QoL of older adults with a master’s degree or above was the highest (Table 3).

For the independent variable “living arrangement” of the elderly, the regression model used “living alone” as the reference group. All dummy variables of living arrangement did not reach the statistical significance level, indicating that the living arrangement of the elderly in Taiwan did not affect the evaluation of quality of life. According to the Survey on the Status of living alone elderly in Statistics Bulletin, there are 42,000 elderly people living alone in Taiwan. The services provided by Taiwan’s Ministry of Health and Welfare for the elderly living alone in 2020 include: home services (29.1%), catering services (35.5%), telephone greetings (19.9%), caring visits (15.3%), escorting to the hospital (0.2%) [32]. Perhaps because the Taiwan government pays special attention to the issue of the elderly living alone, other living arrangement groups do not feel that the quality of life is higher than the elderly living alone.

Older adults who were financially well off were used as the reference group. The QoL of older adults who were roughly enough and financially struggling was poor. The standardized coefficients older adults who were roughly enough and financially struggling were −0.27 and −0.48, respectively, which was statistically significant. Therefore, the poorer the financial situation of older adults, the poorer their QoL (Table 3).

### 3.2. Domain Findings

The standardized estimated coefficients of the four domains of physical health, psychological health, social relationships, and environmental factors were 0.04, 0.31 **, 0.11 **, and 0.16 **, respectively. Thus, the better the psychological health, social relationships, and environmental factors of older adults in Taiwan, the higher their QoL with statistical significantly. The results also indicated that psychological health and environmental factors can improve the QoL of older adults more than social relationships and physical health (Table 3, Figure 1).

### 3.3. Item Findings

After evaluating the domains, this study examined the items of overall QoL as the dependent variable. The items in the four domains of the WHOQOL-BREF were input into four regression models. In the physical health domain, the three items, “activities of daily living”, “dependence on medicinal substances and medical aids”, and “mobility”, were not significant. In the psychological health domain, “thinking, learning, memory, and concentration” was also not significant. In the social relationships domain, all items were statistically significant. In the environmental factors domain, “health and social care: accessibility and quality” and “transport” were not significant (Table 4).

## 4. Discussion

### 4.1. Main Findings

By adopting the psychometric properties questionnaires of WHOQOL-BREF instrument to assess the QoL of elderly people in Swiss, Steinbϋchel has demonstrated that the measurement by a systemic comparison of item-level profiles in specific group would be more informative than the four domains and assessment of QoL would be, to some degree, determined by objective evaluation, such as: health care system, individual expectation level, and economic resources [33]. This study had a random sample of 1078 older adults in Taiwan whose data were collected over landline interviews. This was a relatively large sample size comparing with other similar studies currently. From the above findings, the correlation between 26 items in four domains of the Taiwanese version of WHOQOL-BREF questionnaire and the sociodemographic characteristics, QoL domains, and items from each domain are discussed in the following sections.

#### 4.1.1. Sociodemographic Characteristics

The results of the Sociodemographic Characteristic variables in this study indicated that the women, older-aged, more educated, and more financially well-off elder adult group were predictable having better overall QOL outcome, which was like the findings from several studies previously [34,35,36,37,38].

Although, Lee’s study [39] has found that older men had a higher QoL than older women in five low- and middle-income countries, there were several studies which have indicated that older women may express more emotion and have stronger friendships and social support circles than older men, which may lead to higher QoL scores for older women in Taiwan [40,41] and which coincided with the findings of the studies from Khaje-Bishak and Campos [42,43]. Thus, the overall QoL of older women was relatively higher than that of their male counterparts in this study.

Aligning with the findings from Gobbens in Belgium [37], Jeste determined that older adults who exhibit resilience tend to rate their experience of aging better, in a study involving 1,300 older adults aged 50 to 99 years old living in San Diego [9]. Similar results were noted for the elder age participants especially over 85 years old who also exhibited a relatively high overall QoL score in this study, and which could be as one important factor for predicting the QoL outcome of elders in Taiwan.

Reviewing the studies conducted by Gobbens and Momenabadi [37,44], which indicated that higher education and better financial status could the impact factors for having a healthy physical condition as well as their psychological and environmental QoL by increasing individuals’ self-esteem and empowering them to take part in social activities of elderly people, we had the same results, which revealed that the higher education level, especially in Master’s degree or above, and well off financial situation older adults predicted their higher QoL too.

Liu and Zhou observed a higher overall QoL among older adults in rural China, which may be because they can get care supporting by their children living together in the family [34,35]. We suppose that the proper health insurance and long-term care services offered by the government [45,46] and lower social expectations that family members ought to be responsible for caring their elderly relatives in Taiwan may explain the lack of correlation between living arrangement and overall QoL score.

#### 4.1.2. Relative Domains

Jeste’s study has demonstrated that the self-rated successful aging was associated with better interacting components including physical function, psychological traits, social engagement, and environmental status for the mass community dwelling elder people who live in San Diego, U.S.A. by telephone interview at home [9]. Meanwhile, Tao reported that the social support from medical insurance and pension insurance may produce a “better model” effect on physical health [47]. Besides, Chang’s study indicated that older adults with mild physical challenged may still live with good QoL if their mental or psychological problem was effectively managed [48] and Chammem’s study also indicated that family and social relationships and perception of change in living environment was the more important determinants than was physical health status [49]. Comparing to the research by Goes from the rural area in Portugal which has showed that the physical health domain was considered the best predictor and followed by the psychological health, environment, and social relationships domains [50], the finding by Xia’s study from the urban community in China which has revealed that the predictors of psychological health, social relationships, and environment domains had well acceptable reliability than physical health domain [51].

In recent two decades, Taiwan has made improvement in the QOL on elders by some positive factors. The first one was the “National Health Insurance System” (NHIS) initiation which has provided the effectiveness of improving access to health care and maintaining well quality for physical health care [20]; The second, was conducted two stages of “Long Term Care Ten Years Plane” and “Community Health Creation Program” [46,52] tied the concept of aging in place which has promoted self-action and self-care ability; And the third, “Elderly Welfare Law” (EWL) has initiated several subsidies such as meal delivery, transient caregiver’s services and transportation fee [53]. Similar to the Xia’s finding in China [51], the results in this study which revealed that the psychological health was highest impact factor, the environmental factor and social relationships were the next sequency, and the physical health was the last one and without statistically significant to predict the outcome of QoL in Taiwanese elderly people.

#### 4.1.3. Items

In addition to the domains, we measured the correlation between 26 items within the four domains and the overall QoL score, there were 6 items which revealed without significant correlation to overall QoL outcome and to be exploring the causal factors as below.

Jeste and Hsu and Lin have indicated that physical health conditions, such as resilience and fitness, may lead to more successful aging [9,54,55]. However, the correlation among the E3 (activities of daily living), E4 (dependence on medicinal substances and medical aids), and E15 (mobility) physical health items were not significant. Since 1995, the NHIS has been initiated under a single-payer compulsive social insurance plane. This program has covered annual health examination and provided full grants for those aged over 70 years old [20]. Moreover, with a high satisfaction rate to HHIS (up to 90% in 2019) [45], it would to be believed that the E4 (dependence on medicinal substances and medical aids) is not a key factor for correlation to overall QoL score of older adults in Taiwan because they could easy access health care services including drug acquisition and medical aid from the friendly and successful health insurance services. Moreover, long term care program for elderly adults has been as a priority policy in the past decade and which was tied to the concept of aging in place and community cars services such as: offer supporting services to people over 65 years with limitation on daily living [20], encourage private-sector social welfare groups and government community creation centers initiated by “Health Promotion Administration” to building primary care networks for providing an available, acceptable, and accessible health services to maintain their daily activities and functional mobility of elderly adults [52]. Thus, E3 and E15 may not have a strong correlation with overall QoL score also.

Furthermore, a study by Chi in Taiwan has provided positive evidence supporting the effectiveness of an intervention program in improving psychological status, social participation, and active aging through the Community Health Creation Program. Such programs could aid older adults with cognitive training, emotional awareness, and coping skills during their usual daily activities [56]. The results from Chi may explain why item E7 (thinking, learning, memory, and concentration) in the psychological health domain demonstrated no significant correlation with the overall QoL score. With regard to the government provision of services to older adults, “Long-Term Care Plan 1.0” was promulgated by the Ministry of Health and Welfare in 2007 and its second iteration “Long-term Care Plan 2.0” was introduced in 2016; these plans cover home care, transportation, and community health activities in Taiwan to the present, and takes only 5% to 15% of the reimbursement necessary to pay for the long-term care service fee [46]. There was another subsidy for civic transportation fee to elders for daily use which was provided by local governments under the provision of the “Elderly Welfare Low” (EWL) in Taiwan also. Thus, the elderly adults have little need to worry much about health care access and transportation in their living communities. Consequently, the correlation of items E24 (health and social: accessibility and quality) and E25 (transportation) in the environmental domain with overall QoL was not significant.

### 4.2. Strengths and Limitations

#### 4.2.1. Strengths

Our study reveals that the four domains play the relatively different important role in predicting QoL among elder adult population. Our findings should be interpreted in light of the following limitations. First, this is a cross-sectional study, illustrates that the female gender, age of 85-year-old above, and higher social hierarchy groups (both education and socioeconomic status) tend to have a better overall QoL satisfaction compared to other groups. Second, by ranking sequent, the factors including psychological health, social relationship, and environmental domains are positive correlated to overall QoL score with statistical significance; however, physical health domain factor does not demonstrate significant correlation. The third, among the four domains, there are six items which do not play the significant role in determining overall QoL of Taiwanese elderly, include: (1) activities of daily living, (2) dependence on medicinal substances and medical aids, (3) mobility, (4) thinking/learning/memory/concentration, (5) health and social: accessibility and quality, and (6) transportation. Our investigation concluded that the physical health domain and above six items were not significantly correlative to overall QoL satisfaction of Taiwanese elders due to other contribute factors such as accessible medical services, affordable health insurance system, and community long-term care support.

#### 4.2.2. Limitations

During this study, no large-scale COVID-19 outbreak occurred in Taiwan. The effect of COVID-19 on the QoL of older adults cannot be included in this study. Future research should conduct in-depth investigations on COVID-19′s effect on the QoL of elders. This study primarily collected data through telephone interviews. The responses provided by older adults may be inaccurate due to problems with selective memory and the telescoping effect. Due to time and financial constraints, this study was unable to adopt a longitudinal research approach to accurately determine a causal relationship. Future research should adopt a longitudinal research method with sufficient funds and time. Additionally, this study collected quantitative data, and many in-depth qualitative viewpoints cannot be included. We recommend that future researchers examine both qualitative and quantitative data for more accurate and rigorous results. Finally, due to cultural differences and biases, our findings may not be generalizable for other populations.

## 5. Conclusions

This study conducted telephone questionnaire surveys among the Taiwanese elderly population (aged 65 years and above) by using an adopted-Taiwanese version WHOQOL-BREF stool. This investigation revealed that the four domains play relatively different important roles in predicting QoL among the elder adult population. Moreover, the result of this study also illustrates that not all the items of the WHOQOL-BREF questionnaires can significantly predict QoL of elder adults, and cohort studies should be conducted to demonstrate the consistently reliable result in the future. We also strongly believe that the significant QoL correlative independent variables should be emphasized and considered to be not only applied in formulating long-term care policy by the government, but also for further relevant research purposes.

## Figures and Tables

**Figure 1 ijerph-19-14621-f001:**
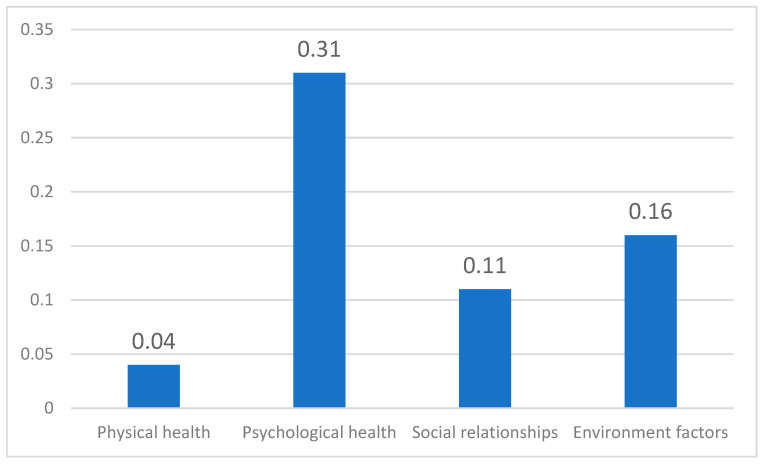
Domain standardized coefficient comparison chart.

**Table 1 ijerph-19-14621-t001:** Reliability and validity analysis of domains.

	Reliability and Validity			
	Lambda Loading	Z-Value	Composite Reliability	Cronbach’s Alpha
**Physical health**			0.85	0.84
E3: Activities of daily living	0.53	− ^1^		
E4: Dependence on medicinal substances and medical aids	0.48	26.8		
E10: Energy and fatigue	0.77	29.2		
E15: Mobility	0.77	29.6		
E16: Pain and discomfort	0.54	26.7		
E17: Sleep and rest	0.82	28.6		
E18: Work capacity	0.77	28.7		
**Psychological health**			0.82	0.81
E5: Positive feelings	0.71	−		
E6: Spirituality/Religion/Personal beliefs	0.68	34.3		
E7: Thinking/learning/memory/concentration	0.73	34.3		
E11: Bodily image and appearance	0.58	29.3		
E19: Self-esteem	0.74	32.3		
E26: Negative feelings	0.46	22		
**Social relationships**			0.70	0.70
E20: Personal relationships	0.71	−		
E21: Sexual activity	0.48	22.4		
E22: Social support	0.61	24.3		
E27: Social Respect	0.64	25.8		
**Environmental factors**			0.79	0.79
E8: Freedom/physical safety/security	0.67	−		
E9: Physical environment (pollution/noise/traffic/climate)	0.47	24.8		
E12: Financial resources	0.62	27.2		
E13: Opportunities for acquiring new information and skills	0.60	27.4		
E14: Participation in and opportunities for recreation/leisure activities	0.54	28.6		
E23: Home environment	0.56	24		
E24: Health and social care: accessibility and quality	0.48	21.1		
E25: Transport	0.54	24		
E28: Food availability	0.43	21.8		

^1^ Fix first facet to 1, there is no Z value in the first facet.

**Table 2 ijerph-19-14621-t002:** Sociodemographic of the subjects.

Sociodemographic Variables	Counts	Percentage (n = 1078)
Gender		
Men	420	39.0%
Women	658	61.0 %
Age		
65–74 years old	633	58.7%
75–84 years old	325	30.1%
85 years old above	120	11.1%
Education Level		
Illiterate	93	8.6%
Elementary school	331	30.7%
Secondary school	113	10.5%
Senior High School	218	20.2%
University	286	26.5%
Master’s degree or above	37	3.4%
Living Arrangement		
Living alone	125	11.6%
Live with spouse	281	26.1%
Living with regular family members	666	61.8%
Other	6	0.6%
Financial situation		
Well off	77	7.1%
Roughly enough	857	79.5%
Struggling	144	13.4%

**Table 3 ijerph-19-14621-t003:** Regression analysis to quality-of-life assessment.

Predictor	Quality of Life Assessment			
	Stand. Estimate	SE	t	*p*
Gender (ref: Men)				
Women	0.12 *	0.04	2.41	0.016
Age (ref: 65–74 years old)				
75–84 years oldd	0.09	0.04	1.57	0.116
85 years old above	0.18 *	0.06	2.25	0.025
Education level (ref: illiterate)				
Elementary school	0.30 **	0.07	3.25	0.001
Secondary school	0.27 *	0.08	2.45	0.015
Senior High School	0.39 **	0.08	3.78	<0.001
University	0.30 **	0.08	2.93	0.003
Master’s degree or above	0.86 **	0.12	5.36	<0.001
Living Arrangement (ref: living alone)				
Live with only two spouses	0.02	0.06	0.22	0.825
Living with regular family members	−0.03	0.06	−0.40	0.69
Other	0.13	0.24	0.40	0.69
Financial situation (ref: well off)				
Roughly enough	−0.27 **	0.07	−2.88	0.004
Struggling	−0.48 **	0.09	−3.98	<0.001
Physical health	0.04	0.01	1.19	0.233
Psychological health	0.31 **	0.01	7.21	<0.001
Social relationships	0.11 **	0.01	3.06	0.002
Environmental Factors	0.16 **	0.01	3.97	<0.001
R^2^	0.412			

Notes: * *p* < 0.05; ** *p* < 0.01.

**Table 4 ijerph-19-14621-t004:** Regression analysis of individual facets to quality-of-life assessment.

Predictor	Quality of Life Assessment			
	Stand. Estimate			
	Model 1	Model 2	Model 3	Model 4
**Physical health**				
E3: Activities of daily living	−0.01			
E4: Dependence on medicinal substances and medical aids	0.05			
E10: Energy and fatigue	0.19 **			
E15: Mobility	0.04			
E16: Pain and discomfort	0.08 *			
E17: Sleep and rest	0.15 **			
E18: Work Capacity	0.14 **			
**Psychological health**				
E5: Positive feelings		0.32 **		
E6: Spirituality/Religion/Personal beliefs		0.13 **		
E7: Thinking/learning/memory/concentration		0.04		
E11: Bodily image and appearance		0.10 **		
E19: Self-esteem		0.14 **		
E26: Negative feelings		0.06 *		
**Social relationships**				
E20: Personal relationships			0.19 **	
E21: Sexual activity			0.12 **	
E22: Social support			0.11 **	
E27: Social Respect			0.24 **	
**Environmental factors**				
E8: Freedom/physical safety/security				0.11 **
E9: Physical environment (pollution/noise/traffic/climate)				0.07 *
E12: Financial resources				0.20 **
E13: Opportunities for acquiring new information and skills				0.16 **
E14: Participation in and opportunities for recreation/leisure activities				0.07 **
E23: Home environment				0.08 **
E24: Health and social care: accessibility and quality				0.03
E25: Transport				0.06
E28: Food availability				0.11 **
R^2^	0.253	0.360	0.240	0.329

Notes: * *p* < 0.05; ** *p* < 0.01.

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
