# Peer review of "Assessing Quality of Life with Community Dwelling Elderly Adults: A Mass Survey in Taiwan"

_ijerph, 2022, doi:10.3390/ijerph192214621_

Round 1

Author Response

Response to Reviewer 1 Comments

Dear Sir,

I would like to thank for your kind and careful review and give us valuable comments on our submitted manuscript. Based on your comments, we have revised the manuscript by actively collected relevant literature with its viewpoint and valuable finding. The following attached statement is the reply under your one-by-one content. Your further guidance is very appreciated, thank you very much.

Point 1: The introduction provides a sound overview of existing research and documents the rationale for the study. Overall, the introduction is well-written, but it would be helpful to provide more explanation about hypothesis 3. Why do authors expect there to be cultural differences? Theoretical background for that would be helpful.

Response 1:

Line 60-66 has added the following text:

In particularly, Lin’s study has pointed out that the Taiwanese cultural perception was having a societal root in Confucianism (respect for the aged and wise) in which has highlighted the cultural value of filial obligation and respect to elders and thus led the long term care policy making based under the concepts of active aging and aging in place to meet the needs of aging population in Taiwan. Meanwhile, the well-established medical care system and economic safety condition in Taiwanese society has been revealed in the past two decades[20].

Line 86-93 has added the following text:

Past studies have found that cultural variables are the most important explanatory factors for QoL in older adults. They found significant differences in QoL between older adults in developed and developing countries [22]. A QoL study of older adults found that individualistic (values referred to as power, achievement, hedonism, stimulation, and self-direction), collectivistic (benevolence, tradition and conformity) or mixed (security and universalism) cultural differences affect evaluation of QoL [23]. Therefore, this study suggests that there will be cultural differences in QoL application among the elderly in Taiwan.

Point 2: Sufficient explanation is needed for study measures (e.g., What was the classification of the financial situation based on? Explain about struggle, roughly enough etc.)

Response 2:

Line 149-158 has added the following text:

The gender variable distinguishes between male and female. The age variable is divided into three categories: 65 to 74 years old, 75 to 84 years old, and 85 years old above. Education level variables are divided into: illiterate, elementary school, secondary school, senior high school, university, master’s degree or above . Living arrangement variables are divided into: living alone, live with spouse, living with regular family members.

The options of the financial situation in this study were modified from the classification of the Survey on the Status of the Elderly in Taiwan: well off, roughly enough, slightly difficult, quite difficult. This study combined “slightly difficult” and “quite difficult” into the “financially struggling” options [26].

Point 3: In table 3, clearly indicate the reference group using “(ref: 65-74 years old; ref: illiterate).”

Response 3:

Added “ref”: to the reference group in Table 3 based on reviewer comments.

Point 4: Further explanation is needed in the interpretation of insignificant correlations.

Response 4:

Line 229-239 has added the following text:

For the independent variable “living arrangement” of the elderly, the regression model used “living alone” as the reference group. All dummy variables of living arrangement did not reach the statistical significance level, indicating that the living arrangement of the elderly in Taiwan did not affect the evaluation of quality of life. According to the Survey on the Status of living alone elderly in Statistics Bulletin, there are 42,000 elderly people living alone in Taiwan. The services provided by Taiwan's Ministry of Health and Welfare for the elderly living alone in 2020 include: home services (29.1%), catering services (35.5%), telephone greetings (19.9%), caring visits (15.3%), escorting to the hospital (0.2%) [32]. Perhaps because the Taiwan government pays special attention to the issue of the elderly living alone, other living arrangement groups do not feel that the quality of life is higher than the elderly living alone.

Point 5: It is necessary to be careful in interpreting some results. For example, the authors state that “In recent decades, Taiwan has made improvement in the QoL on older adults by the positive factors such as: providing proper health care coverage, ensuring economic stability, setting Long-term care projects, and establishing comprehensive social welfare programs.” Evidence is needed to support this statement.

Response 5:

Line 328-339 has revised as the following text:

In recent two decades, Taiwan has made improvement in the QOL on elders by some positive factors. The first one was the “National Health Insurance System” (NHIS) initiation which has provided the effectiveness of improving access to health care and maintaining well quality for physical health care [20] ; The second, was conducted two stages of “Long Term Care Ten Years Plane” and “Community Health Creation Program” [46,52] tied the concept of aging in place which has promoted self-action and self-care ability; And the third, “Elderly Welfare Law” (EWL) has initiated several subsidies such as meal delivery, transient caregiver’s services and transportation fee [53]. Similar to the Xia’s finding in China [51], the results in this study which revealed that the psychological health was highest impact factor, the environmental factor and social relationships were the next sequency, and the physical health was the last one and without statistically significant to predict the outcome of QoL in Taiwanese elderly people.

Line 347-362 has revised as the following text:

Since 1995, the NHIS has been initiated under a single-payer compulsive social insurance plane. This program has covered annual health examination and provided full grants for those aged over 70 years old [20]. Moreover, with a high satisfaction rate to HHIS (up to 90% in 2019) [45], it would to be believed that the E4 (dependence on medicinal substances and medical aids) is not a key factor for correlation to overall QoL score of older adults in Taiwan because they could easy access health care services including drug acquisition and medical aid from the friendly and successful health in-surance services. Moreover, long term care program for elderly adults has been as a priority policy in the past decade and which was tied to the concept of aging in place and community cars services such as: offer supporting services to people over 65 years with limitation on daily living [20], encourage private-sector social welfare groups and government community creation centers initiated by “Health Promotion Administration” to building primary care networks for providing an available, acceptable, and accessible health services to maintain their daily activities and functional mobility of elderly adults [52]. Thus, E3 and E15 may not have a strong correlation with overall QoL score also.

Point 6: It is also necessary to compare it with research applied by WHOQOL in other countries.

Response 6:

Line 269-274 has added the following text:

By adopted the psychometric properties questionnaires of WHOQOL-BREF instrument to assess the QoL of elderly people in Swiss, Steinbüchel has demonstrated that the measurement by a systemic comparison of item-level profiles in specific group would be more informative than the four domains. And assessment of QoL would be in some degree determined by objective evaluation, such as: health care system, individual expectation level, and economic resources [33].

Line 321-327 has added the following text:

Comparing to the research by Goes from the rural area in Portugal which has showed that the physical health domain was considered the best predictor and followed by the psychological health, environment, and social relationships domains [50], the finding by Xia’s study from the urban community in China which has revealed that the predictors of psychological health, social relationships, and environment domains had well acceptable reliability than physical health domain[51].

Reviewer 2 Report

The questionnaire used was adopted and conducted using a randomised telephone interview system from community household elders. Not sure what was done to see if there was a difference with those who did not participate in the study. Also the study was conducted by a survey centre, which was good but was all the interviews standarised?.

Author Response

Response to Reviewer 2 Comments

Dear Sir,

I would like to thank for your kind and careful review and give us valuable comments on our submitted manuscript. Based on your comments, we have revised the manuscript by actively collected relevant literature with its viewpoint and valuable finding. The following attached statement is the reply under your one-by-one content. Your further guidance is very appreciated, thank you very much.

Point 1: The questionnaire used was adopted and conducted using a randomised telephone interview system from community household elders. Not sure what was done to see if there was a difference with those who did not participate in the study.

Response 1:

Line 127-135 has added the following text:

In this study, the first 25% of the samples received and the last 25% received were divided into two groups. In this study, independent sample t-test was performed for physical health, psychological health, social relationships, and environmental factors. The t values of physical health, psychological health, social relationships, and environmental factors were: 0.17, 0.76, 0.66, 0.71. The significant levels of the four domains were 0.862, 0.450, 0.512, and 0.478, which were higher than 0.05. It is confirmed that there is no problem of nonresponse bias in the sample collection of this study. There was no significant difference between those who participated and those who did not.

Point 2: Also the study was conducted by a survey centre, which was good but was all the interviews standarised?

Response 2:

Line 105-114 has added the following text:

This study used standardized interviewing to collect raw data. The advantages of standardized interviewing are that it can reduce bias; increase credibility, reliability and validity; simple, cost-effective and efficient. But standardized interviewing also has some disadvantages: it is difficult to establish a rapport between the interviewer and the participant; the interview is relatively inflexible; the scope of the interview is very limited. The commissioned survey center has some interview methods to reduce the shortcomings of standardized interviewing. They train interviewers to interview in a friendly way, reducing the interviewee's sense of strangeness and stress. They train interviewers to avoid bluntly speaking out the questionnaire and allow respondents some flexibility in answering.

Round 2

Reviewer 1 Report

The revised manuscript is much improved. 

In Table 3, it is not necessary to repeat the reference category continuously if there are more than three categories. For example, in the case of age, it would be good to use it like this:

Age(ref: 65~74 years old)

75~84 years old

85 years old and above